# Chemical and Biological Characterization of *Melaleuca alternifolia* Essential Oil

**DOI:** 10.3390/plants11040558

**Published:** 2022-02-20

**Authors:** Petra Borotová, Lucia Galovičová, Nenad L. Vukovic, Milena Vukic, Eva Tvrdá, Miroslava Kačániová

**Affiliations:** 1AgroBioTech Research Centre, Slovak University of Agriculture, Tr. A. Hlinku 2, 949 76 Nitra, Slovakia; 2Faculty of Biotechnology and Food Sciences, Institute of Applied Biology, Slovak University of Agriculture, Tr. A. Hlinku 2, 949 76 Nitra, Slovakia; eva.tvrda@uniag.sk; 3Faculty of Horticulture and Landscape Engineering, Institute of Horticulture, Slovak University of Agriculture, Tr. A. Hlinku 2, 949 76 Nitra, Slovakia; l.galovicova95@gmail.com; 4Department of Chemistry, Faculty of Science, University of Kragujevac, 34000 Kragujevac, Serbia; nvchem@yahoo.com (N.L.V.); milena.vukic@pmf.kg.ac.rs (M.V.); 5Department of Bioenergy, Food Technology and Microbiology, Institute of Food Technology and Nutrition, University of Rzeszow, 4 Zelwerowicza Str., 35-601 Rzeszow, Poland

**Keywords:** tea tree, essential oil, antioxidant, antimicrobial

## Abstract

The essential oil of *Melaleuca alternifolia*, commonly known as tea tree oil, has many beneficial properties due to its bioactive compounds. The aim of this research was to characterize the tea tree essential oil (TTEO) from Slovakia and its biological properties, which are specific to the chemical composition of essential oil. Gas chromatography/mass spectroscopy revealed that terpinen-4-ol was dominant with a content of 40.3%. γ-Terpinene, 1,8-cineole, and *p*-cymene were identified in contents of 11.7%, 7.0%, and 6.2%, respectively. Antioxidant activity was determined at 41.6% radical inhibition, which was equivalent to 447 μg Trolox to 1 mL sample. Antimicrobial activity was observed by the disk diffusion method against Gram-positive (G^+^), Gram-negative (G^−^) bacteria and against yeasts, where the best antimicrobial activity was against *Enterococcus faecalis* and *Candida albicans* with an inhibition zone of 10.67 mm. The minimum inhibitory concentration showed better susceptibility by G^+^ and G^−^ planktonic cells, while yeast species and biofilm-forming bacteria strains were more resistant. Antibiofilm activity was observed against *Pseudomonas fluorescens* and *Salmonella enterica* by MALDI-TOF, where degradation of the protein spectra after the addition of essential oil was obtained. Good biological properties of tea tree essential oil allow its use in the food industry or in medicine as an antioxidant and antimicrobial agent.

## 1. Introduction

*Melaleuca alternifolia* (tea tree) is a small tree of the family Myrtaceae native to Australia. The tea tree essential oil (TTEO) produced from its leaves can be classified into three major chemotypes: terpinen-4-ol, terpinolene, and 1,8-cineole [1]. Other chemotypes are a combination of dominant and nondominant constituents, with various compositions [2]. The terpinen-4-ol chemotype is dominant and also medicinally more interesting [3]. In medicine, TTEO is used against acne [4], reduces contact dermatitis [5], and improves wound healing [6].

Terpinen-4-ol is a naturally occurring monoterpene alcohol with good antimicrobial properties. It is effective against methicillin-resistant *S. aureus* [7] and fluconazole-resistant *C. albicans* [8]. It can also inhibit proinflammatory mediators [9] and can act as a potential anticancer agent [10]. α-Terpinene and γ-terpinene are monoterpene isomers that naturally occur in various essential oils. Both isomers have strong antioxidant activity and can have an effect on the protection of DNA oxidation [11]. Their next isomer is terpinolene (δ-terpinene), naturally found in herbs, which is frequently used as a flavoring agent. It has high antioxidant capacity and potential antiproliferative properties [12]. 1,8-Cineole, a bicyclic monoterpenoid also known as eucalyptol, is commonly used in the treatment of respiratory diseases [13]. It has good antioxidant and anti-inflammatory properties [14]. Antibiofilm activity against *Streptococcus pyogenes* was also observed [15]. *P*-Cymene belongs to a group of alkylbenzenes. It can be found in more than 100 plant species, and it possesses a broad spectrum of antioxidant, antimicrobial, antiparasitic, and anti-inflammatory activities [16]. Due to bioactive compounds such as terpinen-4-ol, α-terpinolene, α-terpinene, and γ-terpinene, TTEOs have potentially strong antioxidant properties [17]. This natural antioxidant has better properties than synthetic butylated hydroxytoluene, which gives the possibility to prevent food oxidation [18]. The composition of TTEO results in good antimicrobial activity against bacteria, fungal, or viral pathogens [19]. TTEO is able to inhibit respiration and increase membrane permeability, which results in the inhibition of microbial metabolism [20]. Many bacteria are able to grow in the form of biofilm, which is less vulnerable to adverse effects from the outer environment. The inhibition activity of TTEO on pathogenic biofilm was observed against *P. aeruginosa*, *S. aureus*, *C. albicans*, and *V. harveyi* [21]. TTEO is also able to inhibit bacteria in the form of biofilm. TTEO is active against biofilm-forming *S. aureus* [22], *E. coli* [23], or yeast *C. albicans* [24].

*Pseudomonas fluorescens* is a Gram-negative motile bacterium that can form biofilm in soil and water habitats. It can influence food spoilage, water quality, plant diseases, and create nosocomial infections [25]. *P. fluorescens* in the form of biofilm can be found to cause spoilage of milk and dairy products [26]. *Salmonella enterica*, a rod-shaped Gram-negative bacterium, is the main cause of acute foodborne illness and is a source of infection from poultry meat [27]. It can form biofilm even in low temperatures and can cause food spoilage in the refrigerator [28]. The biofilm can be formed on glass and wooden surfaces, which can cause cross-contamination of vegetables [29]. Essential oils have been used for many years in medicine, culinary, and the food industry, due to their beneficial properties. The aim of this research was to characterize EO from Slovakia and its biological properties. The chemical composition of TTEO was determined. Antioxidant and antimicrobial activity were described. This is the first study where the antibiofilm activity of TTEO against *P. fluorescens* and *S. enterica* was determined by MALDI-TOF MS, where molecular changes in protein structure were observed.

## 2. Results

### 2.1. Chemical Composition

In the TTEO, 47 volatile organic compounds were identified, which was 96.5% of the total composition (Table 1, trace components below 0.1% are not listed). The dominant constituent was terpinen-4-ol with 40.3%. The TTEO also contained 11.7% γ-terpinene, 7.0% 1,8-cineole, and 6.2% p-cymene as major components.

### 2.2. Antioxidant Activity

The antioxidant activity measured by DPPH radical scavenging was determined at 41.6% inhibition, which was equivalent to 447 μg Trolox to 1 mL sample.

### 2.3. Antimicrobial Activity Analyzed by Disk Diffusion Method

The highest antimicrobial activity was observed against *E. faecalis* and *C. albicans* with an inhibition zone of 10.67 mm. All other tested microorganisms except *M. luteus* exhibited moderate activity with inhibition zones in the range of 6.00 to 9.33 mm (Table 2). *M. luteus* exhibited the highest resistance with an inhibition zone of 4.67 mm.

### 2.4. Antimicrobial Activity Analyzed by Broth Microdilution Method

The lowest concentration of TTEO with 50% inhibition of microorganisms (MIC 50) was determined for *P. aeruginosa* (10.46 μL/mL), *S. aureus* (11.52 μL/mL), and *S. enterica* (11.82 μL/mL). The lowest MIC 90 values were observed for *P. aeruginosa* (12.32 μL/mL), *S. aureus* (14.26 μL/mL), and *P. fluorescens* (15.46 μL/mL). Generally, G^+^ and G^−^ bacteria seemed to be more susceptible against TTEO compared to yeasts, where MIC 50 reached concentrations above 20 μL/mL. Additionally, biofilm-forming bacterial strains were more resistant to TTEO compared to planktonic bacteria with MIC 50 values of 25.46 μL/mL for *P. fluorescens* and 23.18 μL/mL for *S. enterica* (Table 3).

### 2.5. Analysis of Biofilm Degradation

The effect of TTEO against biofilm-forming bacteria *P. fluorescens* was evaluated with MALDI-TOF MS Biotyper^®^. Spectra of control groups (planktonic cells and biofilm untreated by TTEO) evolved equally (spectra are not shown), so control planktonic cells were used as a control for comparison of molecular changes in biofilm.

Based on the analysis of the mass spectra of the biofilm on individual days (Figure 1A–F), we can observe that the experimental group from the glass surface developed during the experiment very similarly to the control planktonic spectrum up to Day 9. On Days 12 and 14 of the experiment, a change in the mass spectrum was observed, which indicated a change in the protein composition in the biofilm due to long-term exposure to TTEO.

In the experimental group from the wooden surface, we observed a significantly smaller number of peaks with low intensity compared to the control planktonic spectrum from the Day 3 of the experiment, which proves the influence of TTEO on biofilm homeostasis. On Day 14 of the experiment, the number of peaks increased again, which may indicate that the essential oil had a limited duration of action, and the biofilm was able to adapt to adverse conditions.

The constructed dendrogram (Figure 2) showed that the glass experimental group had shorter MSP distances than the control group after Day 9. On Days 12 and 14, the MSP distance of the experimental groups was significantly higher compared to the control group. Based on the findings, we can state that after longer exposure, the inhibitory effect of the essential oil on the structure of the biofilm appeared on the glass surface.

In the wood experimental group, we can observe significantly longer distances of MSP in comparison with the control group after the Day 7 of the experiment. On Day 9 of the experiment, the length of the MSP of the wood experimental group and the control group was the same. On Day 12, a reduction in the difference between the experimental and control groups was visible. On Day 14, there was a shortening of the MSP distance of the experimental group compared to the control. Based on this, we can conclude that the essential oil had an inhibitory effect on the biofilm of *P. fluorescens* up to 7–9 days, and after longer exposure, the biofilm adapted to adverse conditions.

The analyses of mass spectra of *S. enterica* biofilm showed changes in biofilm development from Day 3 after comparing TTEO exposition (Figure 3A). Structural changes were also visible on Days 5, 7, 9, 12, and 14 (Figure 3B–F). The most visible changes in protein structure were observed on Day 5 on the wood surface and on Day 9 on the glass surface.

Based on the constructed dendrogram (Figure 4) on Day 3 of the experiment, the MSP distance of the control group was the same as that of the experimental group on wood, and the experimental group on glass had a shorter distance than the control group. On Day 5, we recorded an increase in the distance of MSP experimental groups compared to the control group. On Day 7, the distances of MSP were shorter. On Day 9, both experimental groups had shorter distances than the control group. On Day 12, the distance between the MSP control group and the experimental group on wood remained significantly higher than the experimental group made of glass. On Day 14, on the other hand, we observed a significantly higher distance between the control group and the experimental group on glass, while the experimental group on wood had a significantly shorter distance from MSP.

The dendrogram showed that development of the *S. enterica* biofilm and protein degradation after TTEO exposure was not as certain as in *P. aeruginosa*, because the length of MSP between the control and experimental groups was not distinguished. While the mass spectra showed changes in protein structure, which was the result of biofilm degradation, the lengths of MPS did not reflect this result. The concentration of TTEO was too low to observe changes in biofilm formation, so the concentration of 0.1% TTEO is not sufficient to degrade biofilm at protein level and needs to be increased.

## 3. Discussion

The TTEO from Slovakia was rich in compounds and classified as a terpinen-4-ol chemotype, which is the most common chemotype. Among the terpinen-4-ol chemotypes, Liao et al. [30] observed the composition of TTEO terpinen-4-ol (40.09%), γ-terpinene (21.85%), α-terpinene (11.34%), α-terpineol (6.91%), and α-pinene (5.86%). In our study, the TTEO contained almost the same amount of the main constituent terpinen-4-ol. Additionally, 1,8-cineole was among the dominant components. Our TTEO also had a lower amount of α-terpinene. Noumi et al. [31] determined their TTEO by the composition of terpinen-4-ol (40.44%), γ-terpinene (19.54%), α-terpinene (7.69%), 1,8-cineole (5.20%), *p*-cymene (4.74%), and α-terpineol (3.31%). TTEO from France had a similar composition to the TTEO from our study. Brun et al. [32] analyzed ten TTEOs, all of which belonged to the terpinen-4-ol chemotype, with contents from approximately 42% to 48%. These TTEOs also contained γ-terpinene from 18% to 25% and α-terpinene from 8% to 12%. Our TTEO had a lower content of all mentioned main components. The composition of TTEO in the study of Kim et al. [18] was terpinen-4-ol (43.2%), γ-terpinene (20.6%), and α-terpinene (9.6%), and Elmi et al. [33] analyzed 41.49% terpinen-4-ol, 20.55% γ-terpinene, and 9.59% α-terpinene in their TTEO sample. Both authors determined a higher percentage of main components than our TTEO. Even though the presence of some constituents is lower than in other studies, our TTEO was comparable to other analyzed TTEOs. All major constituents meet the conditions of ISO norms, except the content of α-terpinene should be over 5% [19]. Various conditions affect the quality of EOs such as geographical location, method of extraction, season of harvesting, or storage conditions.

The method of antioxidant activity is still evolving. Although expression of results has not been standardized, DPPH radical scavenging can validly describe the antioxidant capacity of essential oils [34]. Kim et al. [18] found that 10 μL/mL TTEO methanolic solution approached 80% antioxidant activity by the radical scavenging method, which was higher than our results. Zhang et al. [35] expressed the antioxidant activity as EC_50_ with the concentration of TTEO 48.35 μg/mL and described the TTEO as a potential antioxidant. Jeyakani and Rajalakshmi [36] measured antioxidant activity from 58.52 to 70.41%, which was higher than our antioxidant activity. Zhao et al. [37] determined the antioxidant activity of methanolic solution as approximately 55%. In our study, the antioxidant activity was slightly lower. Shah et al. [38] determined the antioxidant activity in the range of 39.56% to 60.44%, dependent on concentration, which was in accordance with our study. Antioxidant activity is variable and related to chemical composition and can be affected by specific substances. Our TTEO had lower antioxidant properties than the mentioned authors, which can be caused by different chemical compositions. α-Terpinene and α-terpinolene greatly contribute to the antioxidant capacity of TTEO [18], but these components did not dominate in our TTEO.

The antimicrobial activity of TTEO determined by the disk diffusion method was evaluated Puvača et al. [39], who observed antimicrobial activity against *Citrobacter koseri*, *Salmonella* Typhi, and *Escherichia coli* in the range of 13 to 21 mm. This antimicrobial activity was generally stronger than the results from our study. Esmael et al. [40] detected that TTEO was also active against antibiotic-resistant *Staphylococcus aureus* with an inhibition zone of 15.5 mm. The inhibition zone of *S. aureus* from our study is half the size. Melo et al. [41] observed strong inhibition activity against *S. enterica* serovar Typhimurium, *S. enterica* serovar Enteritidis, *E. coli*, *S. aureus*, and *E. faecalis* in the range of 23.43–50.80 mm. In this study, they used double the volume of EO compared to our study. Ergin et al. [42] measured the inhibition zones of six *Candida* species with inhibition zones of 14–42 mm, which were also higher than our inhibition zones. Li et al. [43] observed the inhibition zones of *B. subtilis*, *C. albicans*, *M. luteus E. coli*, *S. enterica* ser. Paratyphi B, *S. aureus*, *S. saprophyticus*, *K. pneumonia*, *S. pyogenes* in the range of 6.60–15.70 mm. *S. aureus* had a lower inhibition zone (6.60 mm) than in our study. On the other hand, *M. luteus* was vulnerable to TTEO with an inhibition zone of 12.9 mm. In our study, *M. luteus* was the most resistant among the tested microorganisms. The authors also stated that *P. aeruginosa* was resistant against TTEO, while in our study, its inhibition zone reached 6 mm. *B. subtilis* and *C. albicans* had higher inhibition zones than in our study (12.6 and 15.7 mm, respectively). Antimicrobial activity is also variable and dependent on factors such as chemical composition. The major active antimicrobial contributor in TTEO is terpinen-4-ol [44,45]. However, minor components also contribute to the antimicrobial efficacy of TTEO [46].

Puvača et al. [39] recorded the MIC of *S.* Typhi, *C. koseri*, and *E. coli* in the range of 2.7 mg/mL to 6.2 mg/mL. They did not determine MIC 50 and MIC 90 values, as in our study, but they claimed good antimicrobial properties, which is in accordance with our study. Ziółkowska-Klinkosz et al. [47] determined the MIC as the concentration that completely inhibited the growth of anaerobic bacteria. They set the concentration in the range of 0.12–0.5 mg/mL which was considered good activity. The authors also declared that their TTEO was more effective against G^+^ bacteria than G^−^. In our study, there were no differences between G^+^ and G^−^ bacteria. However, biofilm-forming G^−^ bacteria and yeasts were less vulnerable to TTEO. Zhang et al. [35] determined the MIC of TTEO against *S. aureus*, *E. coli*, and *P. aeruginosa* at concentrations of 2, 8, and 12 mg/mL, respectively. which was also considered good antimicrobial activity. Hammer et al. [48] tested the inhibition of 16 species of skin bacteria with TTEO; MIC 50 values were 0.25–0.5% *v*/*v*, and MIC 90 values were 0.25–2% *v*/*v*. Bagg et al. [49] observed the inhibition of TTEO against fluconazole and itraconazole-resistant yeast strains. The MIC 50 of TTEO 0.5% *v*/*v* for *C. albicans* and 0.25% *v*/*v* for *C. glabrata* and MIC 90 reached 1% *v*/*v* for both. Types of expression of MIC are also variable, but all authors declared good inhibitory activity of TTEO, which corresponds with our findings.

The influence of TTEO on biofilm formation was observed by Comin et al. [50], who reported the positive influence on inhibition of *P. aeruginosa* after application of TTEO nanoparticles. Kwieciński et al. [51] found out that TTEO can kill all clinical strains of *S. aureus*, both planktonic and biofilm-forming strains. They claimed that 0.5% TTEO killed 99% of bacteria after 60 min. This concentration was more effective but also higher than in our study. Sadekuzzaman et al. [52] found out that TTEO reduced biofilm formation of *E. coli* O157:H7, *L. monocytogenes*, and *Salmonella* spp. at a concentration of 0.1%, which is the same concentration used in our study. Sudjana et al. [53] observed that TTEO inhibited the adhesion of *C. albicans* and its biofilm formation on biotic and abiotic surfaces. Adhesion to polystyrene was significantly reduced at 0.25% TTEO, which was higher than the effective concentration in our study, as we needed only 0.1% TTEO to observe changes in protein structure. Al-Shuneigat [54] found that the TTEO product was able to reduce adherence to polystyrene in the biofilm-forming *Staphylococcus* strain. The reduction was significant at 625 ppm. Song et al. [55] found that TTEO can also effectively reduce the biofilm metabolism of *S. mutans* at a concentration of 0.125%, which was comparable to our study. TTEO showed good growth inhibition against biofilm-forming bacteria, which means that TTEO can potentially be used in the food and pharmaceutical industry to prevent biofilm-forming strains.

Essential oils clearly have an effect on the production of low-molecular-weight protein that can be detected by MALDI-TOF MS. Božik et al. [56] detected ribosomal, membrane-related, cytosol-related, and DNA-related proteins and stress proteins, which changed the expression pattern in *B. subtilis* after essential oil components. Tang et al. [57] suggested that essential oil caused loss of membrane integrity, which led to inhibition of protein and biofilm synthesis at *S. aureus*.

MALDI-TOF MS is able to detect proteins in the mass range of 2–20 kDa, which belongs predominantly to housekeeping proteins, including ribosomal proteins, mitochondrial proteins, cold-shock and heat-shock proteins, and DNA-binding proteins [58,59]. Evaluation of biofilm-forming bacteria is not standard for biofilm formation evaluation. This fast method was described in [60,61] and showed that valid results can be obtained. Few biofilm analyses have been performed by MALDI-TOF, but it seems to be a promising method for the future. Biofilm degradation was observed after exposition by *C. sativum*, *T. serpyllum*, and *T. vulgaris.* EO degradation of low-molecular-protein in the MALDI-TOF MS protein spectra of biofilm-forming bacteria was observed [62,63,64].

## 4. Materials and Methods

The tea tree essential oil was purchased from the Slovak company Hanus s.r.o. (Nitra, Slovakia). The essential oil was extracted by steam distillation of young branches and leaves of *Melaleuca alternifolia.*

### 4.1. Microorganisms

For the antimicrobial analyses, we used four Gram-positive bacteria (*Bacillus subtilis* CCM 1999, *Enterococcus faecalis* CCM 4224, *Micrococcus luteus* CCM 732, and *Staphylococcus aureus* subsp. *aureus* CCM 8223), four Gram-negative bacteria (*Pseudomonas aeruginosa* CCM 3955, *Salmonella enterica* subsp. *enterica* CCM 4420, *Yersinia enterocolitica* CCM 7204, and *Serratia marcescens* CCM 8587), and four yeasts (*Candida albicans* CCM 8261, *Candida glabrata* CCM 8270, *Candida krusei* CCM 8271, and *Candida tropicalis* CCM 8223), which were obtained from the Czech Collection of Microorganisms (Brno, Czech Republic). The biofilm-forming bacterial strains *Pseudomonas fluorescens* and *Salmonella enterica* were used for analyses of antibiofilm activity. *P. fluorescens* was isolated from fish, and *S. enterica* was isolated from chicken meat. Biofilm-forming bacteria were identified by 16S rRNA sequencing and MALDI-TOF MS Biotyper.

### 4.2. Analysis of Chemical Structure

Chemical characterization of TTEO was performed by gas chromatography/mass spectrometry (GC/MS) and gas chromatography (GC-FID) using the Agilent 6890N gas chromatograph (Agilent Technologies, Santa Clara, CA, USA) coupled to a 5975B quadrupole mass spectrometer (Agilent Technologies, Santa Clara, CA, USA). An HP-5MS capillary column (30 m × 0.25 mm × 0.25 μm) was used. The temperature program was set from 60 °C to 150 °C (3 °C/min increasing rate) and 150 °C to 280 °C (5 °C/minincreasing rate) with total run time 60 min. The carrier gas was helium 5.0 with flow rate 1 mL/min. The split/splitless injector temperature was set at 280 °C, and 1 μL EO sample diluted in pentane was injected. Investigated samples were injected in the split mode with split ratio 40.8:1. Electron-impact mass spectrometric data (EI-MS; 70 eV) were acquired in scan mode over the *m*/*z* range 35–550. MS ion source and MS quadrupole temperatures were 230 °C and 150 °C, respectively. GC-FID analyses were performed on an Agilent 6890N gas chromatograph coupled to an FID detector. Column and chromatographic conditions were the same as during GC-MS analysis. FID detector temperature was set at 300 °C. The volatile constituents of EO were identified according to retention indices [65] were compared to reference spectra (Wiley and NIST databases). The retention indices were experimentally determined using the standard method of n-alkanes (C6–C34) retention times, injected under the same chromatographic conditions [66]. The percentages of the identified compounds (higher than 0.1%) were derived from GC peak areas. The measurement was performed in triplicate.

### 4.3. Antioxidant Activity

The radical scavenging of 2,2-diphenyl-1-picrylhydrazyl (DPPH, Sigma Aldrich, Schnelldorf, Germany) was used to measure the antioxidant activity of TTEO. DPPH was dissolved in methanol to concentration 0.025 g/L and was adjusted to absorbance 0.8 at wavelength 515 nm (by Glomax spectrophotometer, Promega Inc., Madison, WI, USA). A 5 μL volume of EO sample was added to 195 μL DPPH solution in a 96-well microplate and was incubated for 30 min in the dark with shaking at 1000 rpm. The percentage of DPPH inhibition was calculated according to the formula (A0 − AA)/A0 × 100, where A0 is the absorbance of DPPH with methanol, and AA is the absorbance of the sample. The standard reference Trolox (Sigma Aldrich, Schnelldorf, Germany) was used for calculation of total antioxidant capacity. Trolox was dissolved in methanol (Uvasol^®^ for spectroscopy, Merck, Darmstadt, Germany) to the concentration range 0–100 µg/mL. Total antioxidant activity was expressed according to the calibration curve as 1 μg Trolox to 1 mL EO sample (TEAC).

### 4.4. Disk Diffusion Method

The disk diffusion method was used for determination of TTEO antimicrobial activity. Bacterial inoculum was cultivated for 24 h on Tryptone soya agar (TSA, Oxoid, Basingstoke, UK) at 37 °C, and yeast inoculum was cultivated on Sabouraud dextrose agar (SDA, Oxoid, Basingstoke, UK) at 25 °C. Microbial culture was adjusted with distilled water to optical density 0.5 McFarland standard (1.5 × 10^8^ CFU/mL). A 100 μL volume of bacterial culture was spread on Mueller–Hinton agar (MHA, Oxoid, Basingstoke, UK), and 100 μL yeast culture was spread on Sabouraud dextrose agar (SDA, Oxoid, Basingstoke, UK). Subsequently, sterile 6 mm disks were saturated with 10 μL TTEO and placed on the microbial suspension, and samples were incubated at 37 °C for bacteria and 25 °C for yeasts for 24 h.

Inhibition zones were measured at three sides from the edge of the filter. The inhibition zone larger than 10 mm was determined as very strong antimicrobial activity, the 10–5 mm inhibition zone was determined as moderate activity, and the 5–1 mm inhibition zone was determined as weak activity. The antibiotics used as control were cefoxitin for G^−^ bacteria, gentamicin for G^+^ bacteria, and fluconazole for yeasts. The method of evaluation of inhibition zones was also the same for biofilm-forming bacteria. Antimicrobial activity was measured in triplicate.

### 4.5. Broth Microdilution Method

Bacterial inoculum was cultivated for 24 h in Mueller–Hinton broth (MHB, Oxoid, Basingstoke, UK) at 37 °C, and yeast inoculum was cultivated on Sabouraud dextrose broth (SDA, Oxoid, Basingstoke, UK) at 25 °C. A 50 μL volume of inoculum with optical density 0.5 McFarland standard was added to a 96-well microplate. A 100 μL volume of TTEO was added to the microbial suspension in final concentrations from 400 μL/mL to 0.2 μL/mL. Samples were mixed incubated for 24 h at 25 °C (yeast cultures) and 37 °C (bacterial cultures) [67].

After 24 h incubation, the biofilm-forming bacteria were dyed with crystal violet. The medium with unattached cells was discarded. Biofilm was washed three times with distilled water. A 200 μL volume of crystal violet (0.1% *w*/*v*) was added, and samples were incubated for 15 min. The solution was discarded, and samples were washed again. The biofilm was solubilized in 200 μL acetic acid (33%). The absorbance was measured at 570 nm [68].

MHB with EO was used as a negative control, and MHB with inoculum was used as a positive control. Absorbance was measured spectrophotometrically at the beginning of the experiment and after 24 h at 570 nm. The analysis was prepared in triplicate.

### 4.6. Analysis of Biofilm Degradation

Degradation of the protein spectra during biofilm development was evaluated in biofilm-forming *P. fluorescens* and *S. enterica.* TTEO addition was evaluated with MALDI-TOF MS Biotyper. A 20 mL volume of MHB was added to 50 mL tubes. A 200 μL volume of biofilm-forming bacteria was added. The tubes also contained the toothpick that simulated the wooden surface and the microscopic slide that simulated the glass surface. TTEO was added to experimental groups in 0.1% final concentration, and control samples were left untreated. The samples were incubated at 37 °C on a shaker with 170 rpm and analyzed on Days 3, 5, 7, 9, 12, and 14. The biofilm samples were taken from glass and wooden surfaces with a sterile cotton swab and were added to the MALDI-TOF target plate. Planktonic cells were taken from 300 µL culture medium. The bacterial suspension was centrifuged for 1 min at 12,000 rpm, and the supernatant was discarded. The pellet was three times washed in 30 μL ultrapure water and centrifuged. After washing, planktonic cells were resuspended, and 1 μL was applied to a target plate.

A 1 μL volume of the α-cyano-4-hydroxycinnamic acid matrix (10 mg/mL) was applied to the dried target plate with samples. MALDI-TOF MicroFlex (Bruker Daltonics) was used for analysis of biofilm protein structure. Spectra were recorded in linear and positive mode with a mass-to-charge ratio range of 200–2000. The protein spectra were obtained by automatic analysis, and the similarities of the spectra in one sample were used to generate the standard global spectrum (MSP). Nineteen MSP were generated from the spectra by MALDI Biotyper 3.0 and were grouped into dendrograms using Euclidean distance [69].

### 4.7. Statistical Data Evaluation

SAS^®^ software was used for data processing. MIC values (concentration that caused 50% and 90% inhibition in bacterial growth) were determined by logit analysis.

## 5. Conclusions

The biological activity and chemical composition of TTEO was evaluated. The TTEO was a terpinen-4-ol chemotype that also contained γ-terpinene, 1,8-cineole, and *p*-cymene. This TTEO can be used as a potential antioxidant. The best antimicrobial activity was observed against *E. faecalis*, and the minimum inhibitory concentration *C. albicans* showed better susceptibility to G^+^ and G^−^ bacteria than yeast and biofilm-forming bacteria. The TTEO was also effective against *P. fluorescens* biofilm formation, while *S. enterica* biofilm needs to be tested with higher concentrations of TTEO.

The TTEO from the Slovak Republic generally has good biological activity, which gives the potential for use in the food industry and medicine to prevent oxidation, inhibit the growth of bacteria, and inhibit the formation of biofilm.

## Figures and Tables

**Figure 1 plants-11-00558-f001:**
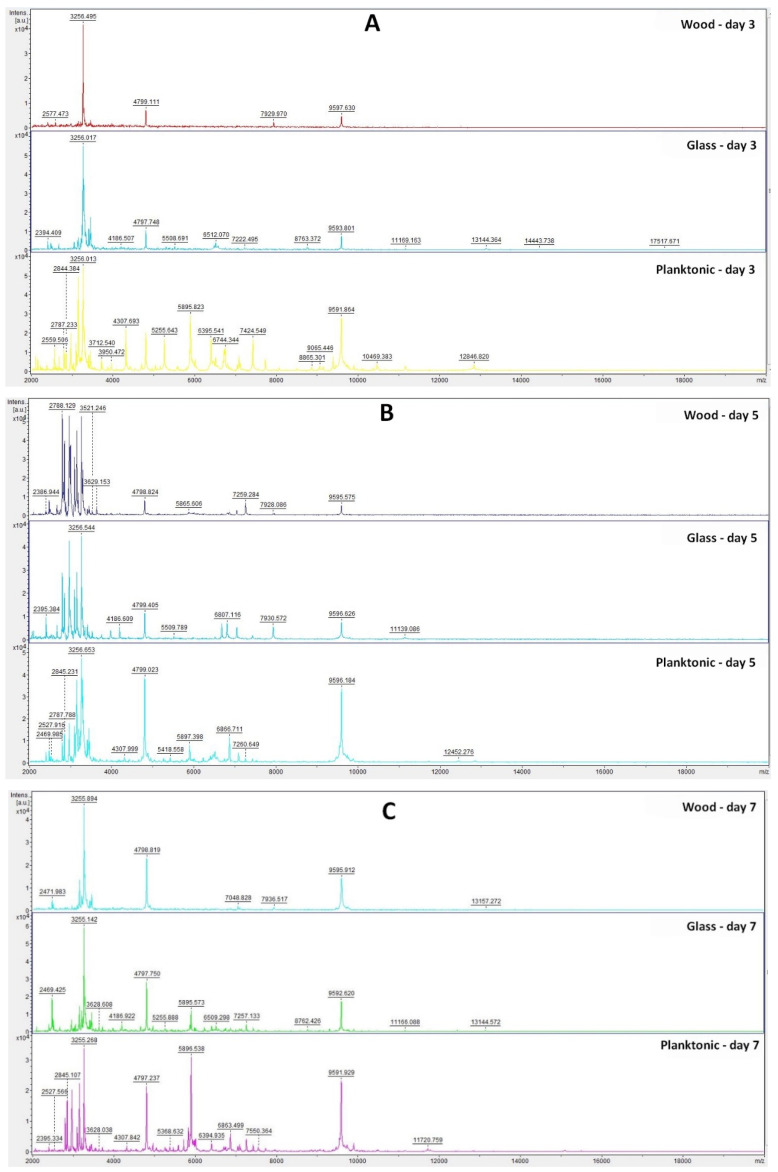
MALDI-TOF mass spectra of *P. fluorescens* biofilm development after TTEO exposition: (**A**) 3rd day; (**B**) 5th day; (**C**) 7th day; (**D**) 9th day; (**E**) 12th day; (**F**) 14th day.

**Figure 2 plants-11-00558-f002:**
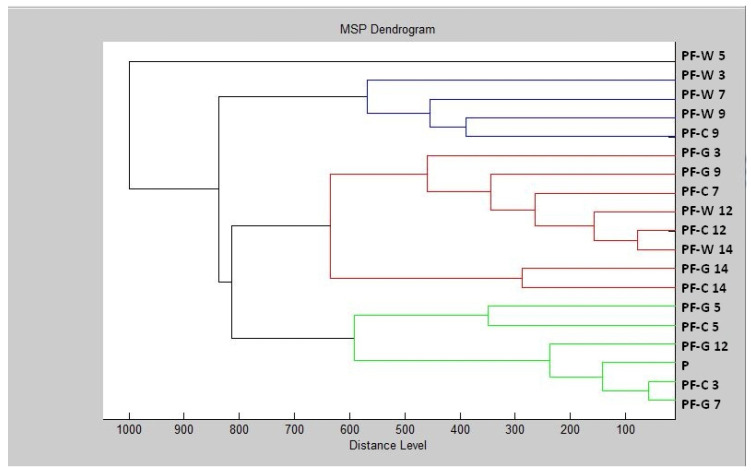
Dendrogram of *P. fluorescens* biofilm progress after TTEO exposition. PF—*P. fluorescens*; C—control; G—glass; W—wood; P—planktonic cells.

**Figure 3 plants-11-00558-f003:**
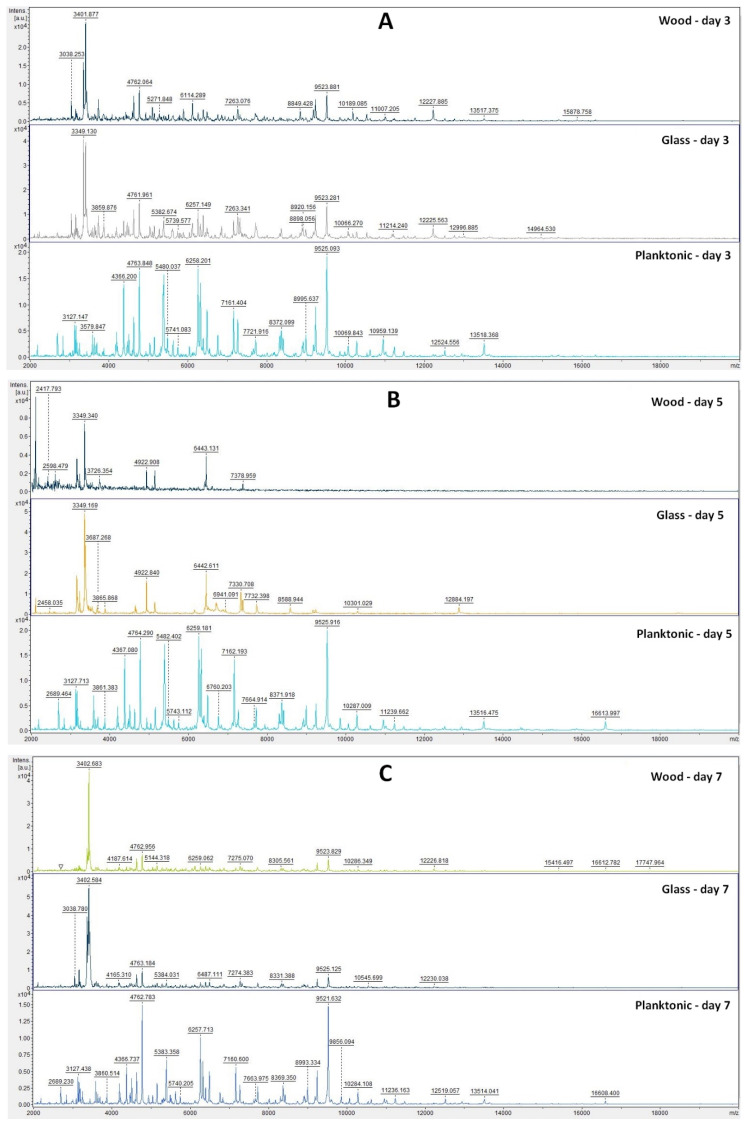
MALDI-TOF mass spectra of *S. enterica* biofilm development after TTEO exposition: (**A**) 3rd day; (**B**) 5th day; (**C**) 7th day; (**D**) 9th day; (**E**) 12th day; (**F**) 14th day.

**Figure 4 plants-11-00558-f004:**
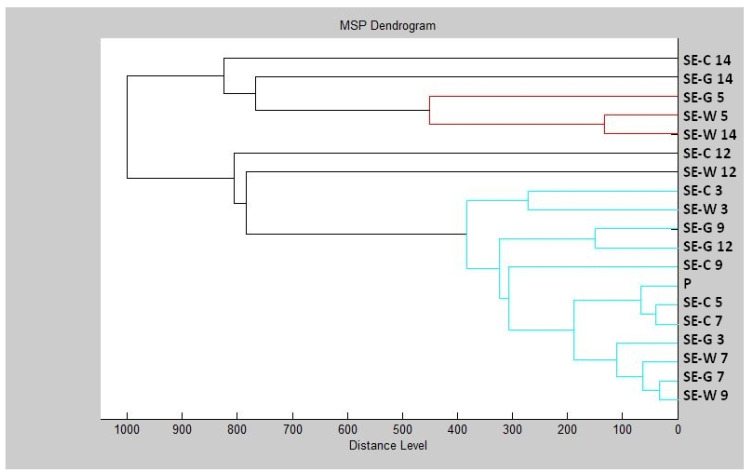
Dendrogram of *S. enterica* biofilm progress after TTEO exposition. SE—*S. enterica*; C—control; G—glass; W—wood; P—planktonic cells.

**Table 1 plants-11-00558-t001:** Chemical composition of TTEO.

Retention Index	Identified Compound	%
1178	Terpinen-4-ol	40.3 ± 0.02
1060	γ-Terpinene	11.7 ± 0.02
1033	1,8-Cineole	7.0 ± 0.01
1023	*p*-Cymene	6.2 ± 0.02
1016	α-Terpinene	3.9 ± 0.01
1189	α-Terpineol	3.9 ± 0.02
938	α-pinene	3.4 ± 0.01
1088	α-Terpinolene	2.2 ± 0.01
1443	Aromadendrene	2.0 ± 0.01
1498	Ledene	1.9 ± 0.01
1525	δ-Cadinene	1.9 ± 0.01
1028	α-Limonene	1.6 ± 0.01
980	β-Pinene	1.2 ± 0.01
992	β-Myrcene	0.8 ± 0.01
1183	*p*-Cymen-8-ol	0.8 ± 0.01
1503	Bicyclogermacrene	0.8 ± 0.01
1530	*cis*-Calamenene	0.8 ± 0.01
1422	(*e*)-Caryophyllene	0.7 ± 0.01
926	α-Thujene	0.6 ± 0.01
1408	α-Gurjunene	0.6 ± 0.01
1098	Linalool	0.5 ± 0.01
1379	α-Copaene	0.5 ± 0.01
1652	α-Eudesmol	0.4 ± 0.01
948	Camphene	0.3 ± 0.01
977	Sabinene	0.3 ± 0.01
1490	β-Selinene	0.3 ± 0.01
1504	α-Muurolene	0.3 ± 0.01
1542	α-Cadinene	0.3 ± 0.01
1004	α-Phellandrene	0.2 ± 0.01
1439	γ-Elemene	0.2 ± 0.01
1456	α-Humulene	0.2 ± 0.01
1577	Spathulenol	0.2 ± 0.01
1593	Viridiflorol	0.2 ± 0.01
1353	α-Cubebene	0.1 ± 0.01
1371	Isoledene	0.1 ± 0.01

**Table 2 plants-11-00558-t002:** Antimicrobial activity of TTEO.

Microorganism	Inhibition Zone	Activity of EO	Control
Gram-positive bacteria			
*Bacillus subtilis*	9.33 ± 1.70	**	31 ± 3.0
*Enterococcus faecalis*	10.67 ± 1.25	***	28 ± 0.5
*Micrococcus luteus*	4.67 ± 0.47	*	26 ± 2.0
*Staphylococcus aureus*	7.33 ± 0.47	**	31 ± 1.0
Gram-negative bacteria			
*Pseudomonas aeruginosa*	6.00 ± 0.82	**	22 ± 1.0
*Yersinia enterocolitica*	6.00 ± 0.82	**	25 ± 2.0
*Salmonella enterica*	7.33 ± 1.25	**	25 ± 1.5
*Serratia marcescens*	6.67 ± 0.94	**	27 ± 2.0
*Pseudomonas fluorescens * biofilm	6.00 ± 0.00	**	26 ± 1.0
*Salmonella enterica* biofilm	6.00 ± 0.82	**	25 ± 1.0
Yeasts			
*Candida albicans*	10.67 ± 1.70	***	25 ± 2.0
*Candida glabrata*	7.67 ± 2.62	**	31 ± 1.5
*Candida krusei*	6.33 ± 0.47	**	31 ± 3.0
*Candida tropicalis*	8.33 ± 1.89	**	31 ± 1.0

* Weak activity (1–5 mm zone); ** moderate activity (5–10 mm zone); *** strong activity (over 10 mm); antibiotics used as control: cefoxitin for G^−^ bacteria, gentamicin for G^+^ bacteria, fluconazole for yeasts.

**Table 3 plants-11-00558-t003:** Minimal inhibitory concentrations of TTEO.

Microorganism	MIC 50 (µL/mL)	MIC 90 (µL/mL)
Gram-positive bacteria		
*Bacillus subtilis*	14.25	18.36
*Enterococcus faecalis*	15.86	18.45
*Micrococcus luteus*	13.58	18.68
*Staphylococcus aureus*	11.52	14.26
Gram-negative bacteria		
*Pseudomonas aeruginosa*	10.46	12.32
*Yersinia enterocolitica*	12.25	15.46
*Salmonella enterica*	11.82	16.36
*Serratia marcescens*	13.45	16.24
*Pseudomonas fluorescens* biofilm	25.46	28.59
*Salmonella enterica* biofilm	23.18	25.43
Yeasts		
*Candida albicans*	22.52	26.76
*Candida glabrata*	24.33	29.85
*Candida krusei*	23.15	26.32
*Candida tropicalis*	21.86	27.46

## Data Availability

Data are contained within the article.

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
