# Peer review of "Chemical and Biological Characterization of Melaleuca alternifolia Essential Oil"

_plants, 2022, doi:10.3390/plants11040558_

Round 1
Reviewer 1 Report
In the manuscript "Chemical and biological characterization of Melaleuca alternifolia essential oil" good biological properties, as antioxidant properties and antimicrobial of tea tree essential oil are shown. This is not a new thing, this plant shows distinct biological properties. My main complaint is the way the biofilm is sampled because it is known that a swab does not give a representative sample for analysis. The question is what the protein profile would be if they used sonication and removed everything from a wooden stick or glass (biomas).
Author Response
In the manuscript "Chemical and biological characterization of Melaleuca alternifolia essential oil" good biological properties, as antioxidant properties and antimicrobial of tea tree essential oil are shown. This is not a new thing, this plant shows distinct biological properties. My main complaint is the way the biofilm is sampled because it is known that a swab does not give a representative sample for analysis. The question is what the protein profile would be if they used sonication and removed everything from a wooden stick or glass (biomas).
Dear reviewer, thank you for your valuable comment. Sonication is also frequently used to break aggregates into smaller pieces for seeding new aggregate growth in laboratory studies of proteins and peptides are associated with various conformational disorders. Sonication affects the change in cell concentration and the protein profile itself should not be changed. Also from the literature, the whole cell analyses by MALDI-TOF MS should be accurate for obtaining of sufficient protein profile.
Reviewer 2 Report
Please, see the attachment.

Author Response
The authors characterized the tea tree essential oil (TTEO) from Slovakia and evaluated the antimicrobial activity against Gram-positive, Gram-negative bacteria and yeasts by disk diffusion method and the Minimum Inhibitory Concentration (MIC). The antibiofilm activity was observed against Pseudomonas fluorescens and Salmonella enterica by MALDI TOF to obtained the degradation of the protein spectra after addition of essential oil.
Some comments to improve the manuscript:
Introduction section:
The introduction section is too short. Please, expand it with information on the main chemical compounds that characterize TTEO and which give it its biological characteristics. Add information on the antibiofilm activity of TTEO and information on the bacteria tested in this study.
Dear reviewer, thank you for your comment, the introduction was expanded with two paragraphs about chemical compounds and about mentioned biofilm forming bacteria.
Results and Discussion section:
Line 72: Please, remove the point after Table 2.
The point was removed.
In Table 2, the authors showed the activity of TTEO against Pseudomonas fluorescens and Salmonella enterica biofilm by Disk Diffusion Method. In the Materials and Methods (section 4.4), the authors don’t describe the Disk Diffusion Method for the evaluation of TTEO activity against Pseudomonas fluorescens and Salmonella enterica biofilm. Please, add this information to the manuscript.
The method for evaluation of disk diffusion method was the same also for biofilm forming bacterial culture. The information was added.
In Table 3, the authors showed the activity of TTEO against Pseudomonas fluorescens and Salmonella enterica biofilm by Agar Microdilution Method. In the Materials and Methods, the authors don’t describe the Agar Microdilution Method for the evaluation of TTEO activity against Pseudomonas fluorescens and Salmonella enterica biofilm. Please, add this information to the manuscript.
The method for evaluation of biofilm-forming bacteria was added to the manuscript.
Line 124: Please, remove the point after 7-9.
The point was removed.
Line 124: Please, change “day” in “days”.
It was changed.
Line 125: Please, add a point after “conditions”.
The period was added to the end of the sentence.
Line 132: Maybe, the most visible changes in protein structure were observed at day 5 on the wood surface and not on the glass surface. Please, verify the data obtained.
Thank you for your comment, the statement was corrected.
Figure 2: The control on day 3 and day 7 is missing in the dendrogram. The glass sample is missing on day 12. Thus, it is difficult to evaluate the results shown in this figure. Please check Figure 2.
The mistakes were made during the renaming in order to unify the samples naming. The marking is correct now according to original MSP dendrogram.
Discussion section
Line 209: Please, write S. enteriditis in italic.
Dear reviewer, to my best knowledge, serovars of Salmonella spp. are written without italic and with capital. Thus, I have added the species of Salmonella.
Line 214: Please, write S. aureus in italic.
aureus had been already in italic, I suppose it was related to S. Paratyphi B. I have also added species name of Salmonella.
Lines 230-231: Maybe, less vulnerable to TTEO?
The statement was corrected.
Materials and Methods section:
Line 270: …”G+ bacteria”. Please, write “Gram-positive bacteria”.
The full name was used instead of abbreviation.
Line 272: …”G- bacteria”. Please, write “Gram-negative bacteria”.
The full name was used instead of abbreviation.
Lines 276-279: Are Pseudomonas fluorescens and Salmonella enterica derived from Collection of Microorganisms or were they isolated from different sources (clinic sample, food samples…)? Please, add this information to the manuscript.
The information has been added.
Section 4.2 (Analysis of Chemical Structure): The analysis were performed in triplicate? Please, add this information to the manuscript.
It was measured in triplicate.
Agar Microdilution Method: The reference to this method is missing.
It was changed as broth microdilution method. Wiegand, I., Hilpert, K. & Hancock, R. Agar and broth dilution methods to determine the minimal inhibitory concentration (MIC) of antimicrobial substances. Nat Protoc 3, 163–175 (2008). https://doi.org/10.1038/nprot.2007.521
There are two sections 4.4. Please check it and change the numbers of the other sections that follow it.
The numbering of the sections has been changed.
Analysis of Biofilm Degradation:
Line 350: Why were the samples incubated on shakers? For biofilm formation, samples must be incubated stationary.
The method was performed according to Pereira et al. 2015 (the reference was added to the method section). They were also shaking the biofilm culture during the growth.
Lines 350-351: Samples were incubated at 37 °C and analyzed on days 3, 5, 7, 9, 12, and 14. During this time, was the medium changed? If the culture medium is not changed during this time, the nutrients decrease too much and the bacterial cells die.
The medium was not changed during the experiment. The amount of broth used was decimal given the duration of the experiments. This method has already been tested in our laboratory using various plant forces and the formation of biofilm was recorded by mass spectrometry.
Please, specify if the biofilm degradation analysis was performed for only some bacterial strains and specify the reason.
The specification was added. As we focus on foodborne pathogens, these two bacteria were chosen. Biofilm production was not recorded in pure bacterial cultures purchased from the Czech collection of microorganisms; therefore they were not tested for biofilm formation.
Why did the authors simulate a wood surface? Why did the authors not simulate a steel surface? Steel is used in both medical and food fields.
Thank you for valuable advice. We tried to simulate various surfaces, and wooden surface is also used in food or dairy industry (either for storage or preparation). This time we have used the wood and glass surface and for next time we will use the steel surface as well.
Conclusion section:
The authors have included results that should not be displayed in this section. Please, modify the conclusion section.
The results were excluded from the conclusion section.
Why did the authors evaluate the antibiofilm activity of TTEO against Pseudomonas fluorescens and Salmonella enterica and not against the bacteria and yeasts used already in a planktonic form? The bacteria and yeasts used in this study are able of forming biofilms.
In our lab, we tried various bacteria to produce biofilm. The one, we have in plaktonic form was not able to grow in form of biofilm during these tests. Also, as I was mentioned earlier, the bacteria which can cause food poisoning were chosen.
Reviewer 3 Report
The paper entitled Chemical and biological characterization of Melaleuca alternifolia essential oil deals with a well-known medicinal plant that was investigated extensively as chemical composition and as an antimicrobial agent.
In this paper, the authors claimed that the plant is from Slovakia, but no information was provided about its geographical origin, condition of cultivation. It seems to be commercialized by the company that they have mentioned in their paper "Slovak company Hanus s.r.o", if this plant is cultivated in Slovakia please show that evidences.
All the microorganism used in this paper are previously screened with tea tree oil, so the novelty of this paper is limited
For Pseudomonas aeruginosa and P. fluorescens, check the following paper
Papadopoulos, Chelsea J., et al. "Susceptibility of pseudomonads to Melaleuca alternifolia (tea tree) oil and components." Journal of Antimicrobial Chemotherapy 58.2 (2006): 449-451.
For the microorganisms Bacillus subtilis, Enterococcus faecalis, Micrococcus luteus and Staphylococcus aureus please check the following paper:
Sharifi‐Rad, Javad, et al. "Plants of the Melaleuca genus as antimicrobial agents: From farm to pharmacy." Phytotherapy Research 31.10 (2017): 1475-1494.
For Candida sp. please check the following paper
However, it is possible to repeat certain antimicrobial tests to confirm the activity but the novelty, in this case, is limited.
In addition, many antibiofilm studies have been done of the same essential oil, but the study of this property for Pseudomonas fluorescens and Salmonella enterica is novel.
Little information was provided about the nature of biofilm composition "which are proteins in your study", please provide more information about their composition and how your experiment using MALDI TOF was useful to detect the protein degradation, what these peaks represent in your experiment?
Many spelling errors are found in the manuscript, please revise this manuscript by a specialized person. For example, write characterization instead of characterisation
Author Response
The paper entitled Chemical and biological characterization of Melaleuca alternifolia essential oil deals with a well-known medicinal plant that was investigated extensively as chemical composition and as an antimicrobial agent.
In this paper, the authors claimed that the plant is from Slovakia, but no information was provided about its geographical origin, condition of cultivation. It seems to be commercialized by the company that they have mentioned in their paper "Slovak company Hanus s.r.o", if this plant is cultivated in Slovakia please show that evidences.
Dear reviewer, thank you for your comment. There is no evidence that Melaleuca alternifolia is grown in Slovak republic, also the statement is not included in the manuscript. That is why only the Slovak company is mentioned.
All the microorganism used in this paper are previously screened with tea tree oil, so the novelty of this paper is limited.
We agree that the work on the same issue has already been published. In our work, biofilm formation of both bacteria was used for the first time using mass spectrometry, and in this we see the novelty of the publication.
However, it is possible to repeat certain antimicrobial tests to confirm the activity but the novelty, in this case, is limited.
Thank you for your comment, we are trying to choose spectra of various microorganisms (G+, G- bacteria, yeasts) to prove the broad antimicrobial ability of essential oils. For next analyses we can choose more precisely the microorganisms which have not been already analyzed. Novelty of the manuscript it is not in antimicrobial activity, we are agree, but how we described previously novelty of manuscript is seems in biofilm formation evaluated with mass spectrometry. In addition, many antibiofilm studies have been done of the same essential oil (it is first study of EOs produced in Slovakia), but the study of this property for Pseudomonas fluorescens and Salmonella enterica is novel.
Little information was provided about the nature of biofilm composition "which are proteins in your study", please provide more information about their composition and how your experiment using MALDI TOF was useful to detect the protein degradation, what these peaks represent in your experiment?
The information have been added to the manuscript.
Many spelling errors are found in the manuscript, please revise this manuscript by a specialized person. For example, write characterization instead of characterization
The spelling errors was caused by use of British English, the text was checked and mistakes were corrected.
Round 2
Reviewer 1 Report
The authors accepted the suggestions and explained the ambiguities and I have no further comments.
Reviewer 2 Report
The authors revised.
Reviewer 3 Report
After checking all the modifications, now the manuscript is improved drastically and can be accepted for publication.
Please make one modification regarding the following sentence line 18
The aim of the research was to characterize the tea tree essential oil (TTEO) from Slovakia
The aim of the research was to characterize the tea tree essential oil (TTEO) from Slavic commercial source.
the same thing in line 76 and 192.